# Effects of atherogenic diet supplemented with fermentable carbohydrates on metabolic responses and plaque formation in coronary arteries using a Saddleback pig model

Lisa Wahl[1,2], Melina Raschke[2,3], Johannes Wittmann[4], Armin Regler[4], Steffen Heelemann[4], Corinna Brandsch[2,3], Gabriele I. Stangl[2,3], Ingrid Vervuert[1,2]*

1 Institute of Animal Nutrition, Nutrition Diseases and Dietetics, Leipzig University, Leipzig, Germany, 2 Competence Cluster of Nutrition and Cardiovascular Health (nutriCARD), Halle-Jena-Leipzig, Germany, 3 Institute of Agricultural and Nutritional Science, Martin Luther University Halle-Wittenberg, Halle (Saale), Germany, 4 Lifespin GmbH, Regensburg, Germany

* ingrid.vervuert@vetmed.uni-leipzig.de

**Data Availability Statement:** All relevant data are within the paper and its Supporting Information files.

## Abstract

Fermentable carbohydrates are gaining interest in the field of human nutrition because of their benefits in obesity-related comorbidities. The aim of this study was to investigate the influence of fermentable carbohydrates, such as pectin and inulin, in an atherogenic diet on metabolic responses and plaque formation in coronary arteries using a Saddleback pig model. Forty-eight healthy pigs aged five months were divided into four feeding groups (n = 10) and one baseline group (n = 8). Three feeding groups received an atherogenic diet (38% crisps, 10% palm fat, and 2% sugar with or without supplementation of 5% pectin or inulin), and one group received a conventional diet over 15 weeks. Feed intake, weight gain, body condition score, and back fat thickness were monitored regularly. Blood and fecal samples were collected monthly to assess the metabolites associated with high cardiovascular risk and fat content, respectively. At the end of 15 weeks, the coronary arteries of the pigs were analyzed for atherosclerotic plaque formation. Independent of supplementation, significant changes were observed in lipid metabolism, such as an increase in triglycerides, bile acids, and cholesterol in serum, in all groups fed atherogenic diets in comparison to the conventional group. Serum metabolome analysis showed differentiation of the feeding groups by diet (atherogenic versus conventional diet) but not by supplementation with pectin or inulin. Cardiovascular lesions were found in all feeding groups and in the baseline group. Supplementation of pectin or inulin in the atherogenic diet had no significant impact on cardiovascular lesion size. Saddleback pigs can develop naturally occurring plaques in coronary arteries. Therefore, this pig model offers potential for further research on the effects of dietary intervention on obesity-related comorbidities, such as cardiovascular lesions, in humans.

**Funding:** This work was supported by the German Federal Ministry of Research and Education (Competence Cluster for Nutrition and Cardiovascular Health (nutriCARD) Halle-Jena-Leipzig, grant number 01EA1808B). The funders had no role in study design, data collection and analysis, decision to publish, or preparation of the manuscript.

**Competing interests:** The authors have declared that no competing interests exist.

## Introduction

The increasing incidence of obesity has become a major global health concern as it is well known that obesity is associated with an elevated risk of cardiovascular disease [1,2]. Various animal models have been developed to investigate the link between obesity and cardiovascular diseases [3–8]. Rodent and lagomorpha models can provide important insights into the influence of metabolism, such as fat metabolism, on the development of atherosclerotic lesions [9–11]. However, rodent and lagomorpha models are limited because atherosclerotic plaques must be artificially induced. In addition, induced plaques occur in arteries different from those in humans [4,11].

In contrast, pig models have attracted interest due to the similarity of their cardiovascular anatomy and physiology with that of humans [12–15]. Moreover, due to similarities in human and pig digestive systems, pig models are advantageous for understanding potential dietary implications in humans [16]. The dietary spectrum of pigs is more similar to that of humans than those of rodents and lagomorpha [13]. Furthermore, the body size of pigs offers advantages in terms of relevant blood and tissue sampling [16]. Previous pig models have used cholesterol with or without the addition of cholic acid to induce the formation of atherosclerotic plaques [17,18]. In Rapacz pigs, familial predisposition to atherosclerosis has been described [19–21]. Furthermore, transgenic pig models have been also used [22]. These studies showed that atherosclerotic plaques developed at similar predilection sites in pigs and humans [13,17–22]. Interestingly, pigs can also spontaneously develop atherosclerotic lesions [23,24]. Using spontaneously occurring lesions in pigs enables the feeding of a typical western-style diet with high amounts of saturated fatty acids, sugar, and salt [25–27].

Recently, various strategies have been developed to counteract obesity-related cardiovascular risk factors. Several studies in humans have shown that fermentable carbohydrates such as pectin and inulin have health benefits in obesity comorbidities [28,29]. In this context the effects of fermentable carbohydrates have been also confirmed by the extensive meta-analysis by Zhou et al. [30].

In the small intestine, pectin and inulin are not digested by endogenous enzymes. Instead, these carbohydrates are available in the large intestine for fermentation by microbiota. The resulting fermentation products are short-chain fatty acids (SCFAs) such as acetate and propionate [31–33]. Consequently, pectin and inulin may lead to a shift in the composition of the microbiota, such as an increase in bifidobacteria [34–39]. Furthermore, a number of positive effects are known from SCFAs, such as immune and inflammation regulation [40]. Studies in humans and pigs have reported a cholesterol-lowering effect of pectin and inulin supplementation [41–43]. Moreover, supplementing high-fat diets with inulin slowed body weight gain in pigs [44]. In addition, inulin supplementation increased satiety by influencing glucagon-like peptides, thereby reducing total energy intake in humans [45,46].

This study aimed to investigate whether fermentable carbohydrates such as pectin and inulin can mitigate cardiovascular risk factors in pigs as a model for obese humans. We hypothesized that SCFA modulation by pectin and inulin will lead to changes in lipid metabolism and may reduce the spontaneous formation of atherosclerotic plaques in pigs fed a western-style diet.

## Materials and methods

### Animals and housing

Forty-eight healthy Saddleback pigs owned by the Institute of Animal Nutrition, Nutrition Diseases and Dietetics, Leipzig University, were included in this study. The animal sample

included 21 female and 27 castrated male pigs from five litters with the same sire. The pigs were aged five months and had a mean (± SD) body weight (BW) of 97.5 ± 9.36 kg. The median [25th / 75th] percentiles of body condition score (BCS) were 3.13 [3.0 / 3.5] out of five, and the mean (± SD) back fat thickness (BFT) was 20.3 ± 2.08 cm at the beginning (t0) of the study. Pigs were housed in the same stable and separated in one pen per group according to the allotted treatment. The ambient temperature was 16–18˚C, and the humidity was 60–75%. The pigs were bedded with wood shavings. Water was provided ad libitum by using an automatic watering system. Animals were adapted to the general experimental environment for at least three weeks. During the adaptation period, pigs were fed the same conventional diet. The project was approved by the Ethics Committee for Animal Rights Protection of the Leipzig District Government (no. TVV 04/20) in accordance with German legislation for animal rights and welfare. Supporting information on the project are provided in a supplementary file (S1 File).

## Study design

The pigs were randomly divided into four feeding groups of ten each (atherogenic diet = AD, atherogenic diet + 5% pectin = ADp, atherogenic diet + 5% inulin = ADi, and conventional diet = CD) and one baseline group (BL; n = 8). After the adaptation period, BL pigs were slaughtered to obtain baseline values of the coronary artery samples. The remaining groups were fed treatment diets for 15 weeks. At the end of the feeding period, pigs were slaughtered for sample collection (January 29 to February 8, 2021).

## Feed management

During the feeding period of 15 weeks, the three groups (AD, ADp, and ADi) received an atherogenic diet that contained high amounts of saturated fatty acids, sugar, and salt. For the ADp group, 5% pectin (agroPECT-A 100/70, agro Food Solution GmbH, Werder, Germany) was added to the atherogenic diet. In the ADi group, 5% inulin (Orafti® IPS; Beneo GmbH, Tienen, Belgium) was added according to literature [36,43,44,54]. The feed was provided ad libitum. The nutrient composition of the different diets is shown in Table 1.

## Health monitoring and morphometric measurements

The health status of each animal was examined in two-day intervals by clinical examination, including evaluations of general behavior, feed and water intake, fecal quality, breathing rate, and body temperature. In addition, blood tests (blood count and chemistry) were performed

**Table 1. Composition of atherogenic and conventional experimental diets.**

| Atherogenic diet | | Conventional diet | |
|---|---|---|---|
| Percentage | Components | Percentage | Components |
| 10 | Wheat | 43.5 | Wheat |
| 18 | Oats | 38 | Oats |
| 17.5 | Soy bean meal | 14 | Soy bean meal |
| 2.5 | Vitamins and Minerals[1] | 2.5 | Vitamins and Minerals[1] |
| 2 | Fiber supplement[2] | 2 | Fiber supplement[2] |
| 38 | Crisps[3] | | |
| 10 | Palm fat[4] | | |
| 2 | Sugar | | |

Data are presented as component percentages. [1]Troumix®M1, Trouw Nutrition Deutschland GmbH, Burgheim, Germany; [2]FaserSpezial 2.0, Trouw Nutrition Deutschland GmbH; [3]By-product of snack industry, feedstuff according to Regulation (EU) No. 575/2011; [4]BEWI-SPRAY® 99L, BEWITAL agri GmbH & Co. KG, Südlohn-Oeding, Germany.

before and after the feeding period. During the feeding period, the daily feed intake per group was recorded. BW was measured weekly using a portable electronic scaling system (Minipond 21; Baumann Waagen und Maschinenbau GmbH, Thiersheim, Germany). BCS was evaluated weekly using a scale from 0 to 5 [47,48]. BFT was obtained monthly by transcutaneous ultrasound measurements (Portable Ultrasonic Diagnostic System A6V, SonoScape Co., Shenzhen, China) at six measurement points according to the ABC-6-method [49].

## Blood sampling

Blood samples were collected at the beginning of the study (t0; October 8–13, 2020) via a single puncture of the left or right jugular vein. Follow-up blood samples of the four feeding groups were collected after one (t1; November 11, 2020), two (t2; December 17, 2020), and three months (t3; January 21, 2021) of experimental diet feeding. For blood chemistry and metabolome analyses, serum tubes (Monovette® Z; Sarstedt AG & Co. KG, Nümbrecht, Germany) containing a coagulation activator was used. Blood count tubes containing EDTA (Monovette® K3E (1.6 mg EDTA/mL); Sarstedt AG & Co. KG) were analyzed immediately after sampling. Serum tubes were centrifuged after 30 min of clotting at room temperature, and the serum was frozen in multiple aliquots of 1 mL at −80°C until analysis.

## Fecal sampling

Fecal samples were collected at the same time points (t0–t3) as the blood samples. Rectal feces were collected from each animal. Pooled fecal samples were prepared for each feeding group and analyzed directly.

## Slaughtering and sampling of coronary arteries

The pigs were stunned using electrical stunning equipment (TGB 200; Hubert Haas, Neuler, Germany). Blood was withdrawn immediately after electrical stunning by severing the brachiocephalic trunk and jugular vein. Pigs were slaughtered in accordance with European and German law [Council Regulation (EC) No 1099/2009 of 24 September 2009, Tierschutz-Schlachtverordnung, § 4 Tierschutzgesetz]. After slaughter, the hearts were separated in toto for subsequent sampling and washed with isotonic saline (NaCl, Carl Roth GmbH + Co. KG, Karlsruhe, Germany). The left anterior descending branch of the left coronary artery (LAD) was manually flushed with isotonic saline, and four 2 mm segments of each animal were removed 3 mm after the bifurcation of the left main stem of the coronary artery, where the artery got divided into the left circumflex and the LAD. The 1st and 3rd LAD segments per animal were shock-frozen in liquid nitrogen and stored at -80°C until frozen sections were prepared. The 2nd and 4th LAD segments were fixed in 10% neutral-buffered formalin (Sigma-Aldrich, St. Louis, USA) for at least two days to prepare polyethylene glycol sections.

## Analysis

**Diets.** All diets were analyzed for crude nutrient and fiber fractions. Dry matter (DM) was determined after oven-drying (103°C). Crude nutrient contents were assayed using the Weende system [50]. Crude fiber (CF), ash free neutral detergent fiber (NDF), and ash free acid detergent fiber (ADF) were analyzed using ANKOM A220® (ANKOM Technology, Salzwedel, Germany) according to Van Soest, Robertson, & Lewis (1991) [51]. The content of nitrogen-free extracts (NFE) was calculated as follows:

NFE = DM – (CA + CP + CL + CF); (CA = crude ash, CP = crude protein, CL = crude lipid). Starch was quantified polarimetrically (VDLUFA III, 7.2.1 Erg. 2012), and sugar content

**Table 2. Nutrients and calculated metabolizable energy levels of experimental diets.**

| Crude nutrients in DM (%) | AD | AD + 5 % pectin | AD + 5 % inulin | CD |
|---|---|---|---|---|
| CA | 5.27 | 5.42 | 5.02 | 4.77 |
| CL | 27.2 | 25.7 | 25.2 | 3.40 |
| CP | 15.1 | 14.7 | 14.2 | 19.0 |
| CF | 7.22 | 7.94 | 7.34 | 7.13 |
| NDF | 11.7 | 10.7 | 10.7 | 21.7 |
| ADF | 5.60 | 5.56 | 5.45 | 9.46 |
| NFE | 39.2 | 39.9 | 41.9 | 55.4 |
| Starch | 34.4 | 34.3 | 32.4 | 45.5 |
| Sugar | 6.71 | 6.16 | 10.8 | 3.35 |
| Sodium | 0.34 | 0.35 | 0.34 | 0.14 |
| **ME (MJ/kg DM) [51]** | **17.8** | **17.2** | **17.3** | **14.1** |

The data are presented as percentages of DM. DM, dry matter; AD, atherogenic diet; CD, conventional diet; CA, crude ash; CL, crude lipid; CP, crude protein; CF, crude fiber; NDF, ash-free neutral detergent fiber; ADF, ash-free acid detergent fiber; NFE, nitrogen-free extracts; ME, metabolizable energy.

was determined gravimetrically (VDLUFA III, 7.1.3, 1976; LKS mbH, Lichtenwalde, Germany). Sodium was analyzed using inductively coupled plasma optical emission spectrometry (ICP-OES) according to DIN EN ISO 11885:2009–09 (LKS mbH). The analyzed values were used to calculate the metabolizable energy (ME) content in DM [52]: ME (MJ/kg DM) = $0.021503 \times CP$ (g/kg) + $0.032497 \times CL$ (g/kg)– $0.021071 \times CF$ (g/kg) + $0.016309 \times$ starch (g/kg) + $0.014701 \times$ organic residue (g/kg); organic residue = organic fraction–(CP + CL + starch + CF). The nutrient composition of each diet is shown in **Table 2**.

**Serum parameters of liver and lipid metabolism.** Serum triglyceride (TG), bile acids (BA), hepatic triglyceride lipase (LIPC), cholesterol (CHOL), alkaline phosphatase (ALP), aspartate aminotransferase (AST), gamma-glutamyl transferase (GGT), lactate dehydrogenase (LDH) and amylase (AMYL) levels were analyzed using an automated chemistry analyzer (Roche Cobas C311, Roche Diagnostic GmbH, Mannheim, Germany). Blood counts were evaluated in all pigs before and after the 15-week feeding period using an ADVIA 120 (Siemens Healtheneers, Dreieich, Germany).

**[1]H Nuclear magnetic resonance (NMR) spectroscopy of serum samples.** Metabolomic analysis was performed using NMR spectroscopy (Lifespin GmbH, Regensburg, Germany). Serum samples were thawed at room temperature and inverted five-fold; 350 μL serum and 350 μL aqueous buffer ($H_2O$ p.A., 0.1 g/L $NaN_3$, 0.067 mol/L $Na_2HPO_4$, 0.033 mol/L $NaH_2PO_4$ (pH-value: 7.15 ± 0.05), 5% $D_2O$, 6 mM pyrazine as an internal standard for quantification) were mixed. Then, 600 μL of the mixture was transferred to a 5 mm NMR tube (Bruker Corporation, Billerica, USA). The samples were kept at 4°C prior to measurement and analyzed within 24 h.

NMR spectra of the serum were recorded at 310 K using an AVANCE NEO spectrometer (Bruker Corporation) operating at a proton frequency of 600 MHz. Spectra were recorded using a NOESY pulse sequence with water presaturation "noesygppr1d" in Bruker notation. For each sample, 16 subsequent scans were collected with a 10 s relaxation delay, an acquisition time of 2.75 s, 96 k data points, and a spectral width of 30 ppm.

The spectra were processed, and 102 blood serum metabolites were quantified using Lifespin's proprietary profiling software version 1.4 (Lifespin GmbH). The concentration of any

NMR-measured metabolite was obtained as a signal integral of non-overlapping resonances or a cluster of partly overlapping resonances. The metabolite resonances were identified according to chemical shift assignments using Lifespin´s proprietary substance reference database (Lifespin GmbH).

**Fat content of the pooled fecal samples.**   The amount of CL was determined in pooled fecal samples from the four feeding groups (AD, ADp, ADi, and CD) and at each sampling point (t0–t3). The analysis of CL in feces was performed analogously to the analysis of CL in the diets.

**Histological examination of the LAD.**   To prepare frozen artery sections, a freezing microtome (CM 1859, Leica Biosystems Nussloch GmbH, Nussloch, Germany, -25˚C) was used. The segments were aligned in embedding medium (Epredia™ Neg-50™ Frozen Section Medium, Thermo Fisher Scientific, Waltham, USA) and fixed with a freezing spray (Shandon Enviro-Tech Freezing Spray, ThermoFisher Scientific). Cross sections with a layer thickness of 7 μm were prepared.

To prepare PEG sections according to Mulisch and Welsch (2015) [53], the segments were first dehydrated and then embedded in polyethylene glycol (PEG 1500, Merck KGaA, Darmstadt, Germany). After embedding, sections with a layer thickness of 4 μm were prepared with a rotary microtome (HM 335 E, Microm International GmbH, Walldorf, Germany). The arterial sections were stained by Movat's Pentachrome to quantify the size of the vascular lesions. Based on the histological data, the ratio between the plaque and vessel area was calculated. Additionally, LAD sections were stained with Oil red to visualize lipids and with Von Kossa staining to detected vascular calcification. An immunohistochemical analysis was performed to identify macrophages using a specific mouse anti pig macrophages antibody (MCA2317GA, BIO-RAD, Hercules, USA). To detected smooth muscle cells a mouse anti human actin alpha (smooth muscle) antibody (MCA5781GA, BIO-RAD), previously applied for porcine tissue was used.

## Statistics

Data analysis was performed using commercial statistical software (SPSS Statistics 27.0, IBM, New York, USA and STATISTICA 14.0, TIBCO, Palo Alto, USA). All data were checked for normal distribution using the Shapiro–Wilk test and for variance homogeneity using the Levene test. The datasets of BWs, BCS, BFT, CHOL and GGT were normally distributed and homogeneous. Repeated measures ANOVAs were used for these parameters, and as a post hoc test, the Tukey HSD test was applied. As the data sets were not normally distributed, the Kruskal–Wallis test with Bonferroni correction was performed (TG, BA, LIPC, AST, LDH, AMYL, ALP and plaque size). The chi-square test was used to evaluate plaque frequency per group. The level of significance was set at $P < 0.05$. Data are shown as medians and [25th / 75th] percentiles for TG, BA, CHOL, and plaque size or mean values ± standard deviation (mean ± SD) for BW, BCS, and BFT.

Further statistical analyses were conducted using R (R version 4.0.2. 2021, R Core Team). Principal component analysis (PCA) was performed on the serum NMR data to show that the groups could be separated by metabolites. In addition, NMR data were analyzed using partial least squares-discriminant analysis (PLS-DA). A default seven-fold cross-validation strategy and a permutation test (200 permutations) were performed to avoid overfitting PLS-DA.

## Results

### Feed intake and morphometric measurements

The total feed intake was similar between the different feeding groups (3680 ± 360 kg). All pigs showed significant increases in BW, BCS, and BFT. However, there were no significant group

**Table 3. Increases in BW, BCS, and BFT in the four feeding groups during the observation period.**

| Parameter in (%) | Group AD | Group ADp | Group ADi | Group CD |
|---|---|---|---|---|
| BW | 61.2 ± 16.3 | 58.1 ± 12.0 | 62.1 ± 11.8 | 58.9 ± 8.60 |
| BSC | 32.1 ± 13.9 | 33.8 ± 6.30 | 33.6 ± 11.2 | 33.0 ± 12.3 |
| BFT | 110 ± 41.9 | 90.9 ± 45.3 | 109 ± 53.1 | 82.3 ± 44.7 |

Data are presented as mean ± SD. Significant differences between feeding groups (n = 10) were identified by P values ≤ 0.05, using repeated measures ANOVA with Tukey's HSD. AD, group fed atherogenic diet; ADp, group fed atherogenic diet + pectin; ADi, group fed atherogenic diet + inulin; CD, group fed conventional diet; BW, body weight; BCS, body condition score; BFT, back fat thickness. The observation period was 15 weeks.

differences in BW gain (P = 0.82), BCS (P = 0.99), or BFT (P = 0.42) (**Table 3**). Original data of BW, BCS and BFT are provided in a supplementary table (**S1 Table**).

### Fecal fat content

The CL content in the DM of pooled fecal samples (**Table 4**) increased after diet change (t1) in the groups fed atherogenic diets (AD, ADp, and ADi).

### Serum parameters of liver and lipid metabolism

Serum parameters were significantly different between the feeding groups (**Fig 1** and **Table 5A–5D**). For example, significantly higher concentrations of TGs and BAs were determined at times t1 and t2 in pigs in the AD, ADp, and ADi groups than in the CD group. The same significant group difference between the pigs fed the atherogenic diet and the pigs fed the conventional diet was observed for LIPC at time t1. The concentration of CHOL was significantly higher in the AD group than in the CD group at time t1. The other parameters (ALP, AST, GGT, LDH and AMYL) without treatment related effects are provided in a supplementary table (**S2 Table**). Original data of serum analysis are also provided in a supplementary table (**S3 Table**).

### Multivariate data analyses of serum $^1$H NMR data

First, unsupervised PCA was performed on the centered and scaled to unit variance serum NMR data to determine the general structure of each dataset. The PCA score plots (**Fig 2**) showed that the metabolite profiles of the AD groups (AD, ADp, and ADi) differed significantly from the CD group and could be clearly distinguished at t1, t2, and t3 but not at t0.

Additionally, supervised PLS-DA with mean-centered and unit variance NMR data was performed. A permutation test (200 permutations) was performed to confirm the reliability of the models. PLS-DA score plots (**Fig 3**) revealed a separation among the four feeding groups

**Table 4. Crude lipid content in pooled fecal samples of each dietary group (AD, ADp, ADi, CD) at each sampling point (t0–t3).**

| CL in DM (%) | Groups | t0 (start) | t1 (1 month) | t2 (2 months) | t3 (3 months) |
|---|---|---|---|---|---|
| | AD | 6.27 | 24.6 | 17.5 | 21.1 |
| | ADp | 6.44 | 24.2 | 25.3 | 17.9 |
| | ADi | 7.21 | 21.8 | 18.5 | 20.9 |
| | CD | 5.98 | 4.10 | 4.81 | 4.14 |

Data are presented as percentages (in DM) for one pool of ten animals per group. CL, crude lipid; DM, dry matter; AD, group fed an atherogenic diet (n = 10); ADp, group fed an atherogenic diet + pectin (n = 10); ADi, group fed an atherogenic diet + inulin (n = 10); CD, group fed a conventional diet (n = 10).

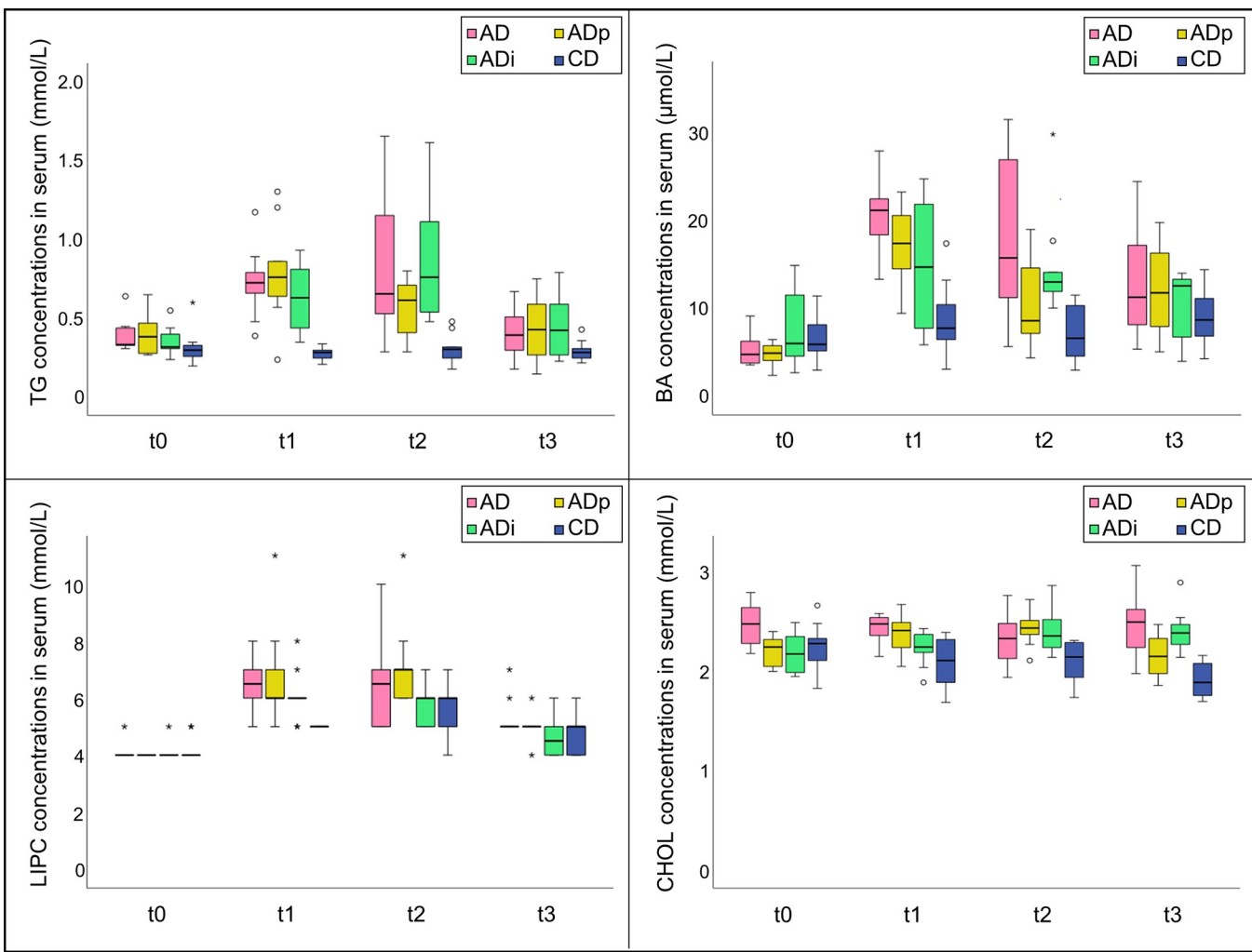

**Fig 1. Triglycerides, bile acids, hepatic triglyceride lipase, and cholesterol concentrations in serum.** Box plots show the concentration of TG, BA, LIPC and CHOL at the time point before the feed change (t0) and time points 1–3 months after the feed changes (t1–t3) for all feeding groups (AD, ADp, ADi, CD). TG, triglycerides; LIPC, hepatic triglyceride lipase; BA, bile acids; CHOL, cholesterol; AD, group fed atherogenic diet (n = 10); ADp, group fed atherogenic diet + pectin (n = 10); ADi, group fed atherogenic diet + inulin (n = 10); CD, group fed conventional diet (n = 10); °*, outliers and extreme values.

for t1 $(Q^2 = 0.29, R^2X = 0.38, R^2Y = 0.4, pQ^2 = 0.005)$ and t2 $(Q^2 = 0.59, R^2X = 0.48, R^2Y = 0.81, pQ^2 = 0.005)$. At t3 $(Q^2 = 0.9, R^2X = 0.41, R^2Y = 0.97, pQ^2 = 0.005)$, only ADs (including AD, ADp and ADi) and CD could be distinguished. At time point t0, a visual separation is possible; however, it can be classified as overfitting based on the performance indicators $(Q^2 = -0.02, pQ^2 = 0.09)$.

In total, 102 metabolites were identified in this study. In particular, metabolites related to fat metabolism, including cholesterol, fatty acid residues, and saturated carbon chain elements, were more abundant in the AD, ADp, and ADi groups than in the CD group.

## Plaque formation

Moderate plaque formation in the LAD was found in the baseline group and in the four feeding groups (**Fig 4**). All plaques detected in the LAD sections represent very early lesion stages of atherosclerosis without calcification, high amounts of lipids, invasion of macrophages or differences in stainable smooth muscle cells. In the ADp group, significantly more animals had

**Table 5. a: Triglycerides concentrations in serum (mmol/L) of all groups at each sampling point (t0–t3).** b: Bile acids concentrations in serum (μmol/L) of all groups at each sampling point (t0–t3). c: Hepatic triglyceride lipase concentrations in serum (mmol/L) of all groups at each sampling point (t0–t3). d: Cholesterol concentrations in serum (mmol/L) of all groups at each sampling point (t0–t3).

| Groups | t0 | t1 | t2 | t3 |
|---|---|---|---|---|
| BL | 0.38 [0.28 / 0.49] | n/a | n/a | n/a |
| AD | 0.34 [0.33 / 0.44] | 0.73[#, a] [0.62 / 0.82] | 0.66[#, a] [0.52 / 1.21] | 0.40 [0.29 / 0.53] |
| ADp | 0.39 [0.28 / 0.50] | 0.76[#, a] [0.62 / 0.95] | 0.62[#, ab] [0.38 / 0.71] | 0.43 [0.27 / 0.60] |
| Adi | 0.32 [0.30 / 0.41] | 0.63[#■, a] [0.44 / 0.82] | 0.76[■, a] [0.54 / 1.17] | 0.43[#] [0.26 / 0.63] |
| CD | 0.30 [0.25 / 0.34] | 0.29[b] [0.25 / 0.30] | 0.31[b] [0.24 / 0.35] | 0.29 [0.25 / 0.32] |

Triglycerides concentrations are presented as medians and [25th / 75th] percentiles in mmol/L. [#■]Different symbols indicate significant differences within a row (time-point differences in a group). [ab]Lowercase letters indicate significant effects within a column (group differences at one timepoint). Significant differences are identified by P values ≤ 0.05 using Kruskal–Wallis test with Bonferroni correction. BL, baseline group (n = 8); AD, group fed atherogenic diet (n = 10); ADp, group fed atherogenic diet + pectin (n = 10); ADi, group fed atherogenic diet + inulin (n = 10); CD, group fed conventional diet (n = 10); n/a, not available.

plaques (10/10 animals affected) than in the AD group (5/10 animals affected, **Table 6**). Plaque sizes did not differ significantly between the groups (P = 0.33, Kruskal-Wallis test with Bonferroni correction). Additionally, no association was found between the plaque frequency per group and plaque size. The smallest plaque was observed in the baseline group. Original data of plaque formation are provided in a supplementary table (**S4 Table**).

## Discussion

In our study, we developed a pig model under well-defined feeding and housing conditions as an atherogenic dietary model for humans. Dietary modeling using pigs is advantageous

| Groups | t0 | t1 | t2 | t3 |
|---|---|---|---|---|
| BL | 5.25 [3.83 / 5.70] | n/a | n/a | n/a |
| AD | 4.70 [3.68 / 6.40] | 21.2[#, a] [17.2 / 23.5] | 15.8[#, a] [11.0 / 27.9] | 11.3[#] [7.73 / 17.6] |
| ADp | 4.85 [3.88 / 5.70] | 17.4[#, a] [14.0/ 21.2] | 8.55[#, ab] [7.08 / 15.3] | 11.8[#] [7.60 / 16.4] |
| ADi | 5.95 [4.25 / 11.8] | 14.7[#, ab] [7.48 / 22.3] | 13.0[#, a] [11.7 / 15.0] | 12.6[#] [6.30 / 13.5] |
| CD | 5.85 [5.00 / 8.30] | 7.70[b] [6.38 / 11.1] | 6.55[b] [4.38 / 10.5] | 8.65 [6.25 / 11.3] |

Bile acids concentrations are presented as medians and [25th / 75th] percentiles in μmol/L. [#■]Different symbols indicate significant differences within a row (time-point differences in a group). [ab]Lowercase letters indicate significant effects within a column (group differences at one timepoint). Significant differences are identified by P values ≤ 0.05 using Kruskal–Wallis test with Bonferroni correction. BL, baseline group (n = 8); AD, group fed atherogenic diet (n = 10); ADp, group fed atherogenic diet + pectin (n = 10); ADi, group fed atherogenic diet + inulin (n = 10); CD, group fed conventional diet (n = 10); n/a, not available.

| Groups | t0 | t1 | t2 | t3 |
|---|---|---|---|---|
| BL | 5.00 [4.25 / 5.00] | n/a | n/a | n/a |
| AD | 4.00 [4.00 / 4.00] | 6.50[#, a] [6.00 / 7.00] | 6.50[#, ab] [5.00 / 7.50] | 5.00[#] [5.00 / 5.25] |
| ADp | 4.00 [4.00 / 4.00] | 6.00[#■, a] [6.00 / 7.25] | 7.00[■, a] [6.00 / 7.25] | 5.00[#] [5.00 / 5.00] |
| ADi | 4.00 [4.00 / 4.00] | 6.00[#, a] [5.75 / 6.25] | 6.00[#, ab] [5.00 / 6.25] | 4.50 [4.00 / 5.00] |
| CD | 4.00 [4.00 / 4.25] | 5.00[#, b] [5.00 / 5.00] | 6.00[#, b] [5.00 / 6.00] | 5.00[#] [4.00 / 5.25] |

Hepatic triglyceride lipase concentrations are presented as medians and [25th / 75th] percentiles in mmol/L. [#■]Different symbols indicate significant differences within a row (time-point differences in a group). [ab]Lowercase letters indicate significant effects within a column (group differences at one timepoint). Significant differences are identified by P values ≤ 0.05 using Kruskal–Wallis test with Bonferroni correction. BL, baseline group (n = 8); AD, group fed atherogenic diet (n = 10); ADp, group fed atherogenic diet + pectin (n = 10); ADi, group fed atherogenic diet + inulin (n = 10); CD, group fed conventional diet (n = 10); n/a, not available.

compared to that using other laboratory animals due to factors such as similarities in food spectra, digestive systems, and the manifestation of coronary heart disease. The atherogenic diet used in this study was comparable to a typical western-style human diet. Analyses of several parameters relevant to evaluating cardiovascular and cardiometabolic health were performed. We hypothesized that supplementation of an atherogenic diet with two different fermentable carbohydrates, pectin and inulin, has beneficial effects on metabolism and plaque formation.

As expected, feed intake was high in all feeding groups, and the amount of feed intake was comparable between the groups. Significant increases in BW, BCS, and BFT were observed in all the groups. However, despite differences in energy content between atherogenic diets (17.2–17.8 MJ ME/kg DM) and the conventional diet (14.1 MJ ME/kg DM) increase in BW,

| Groups | t0 | t1 | t2 | t3 |
|---|---|---|---|---|
| BL | 2.37 [2.18 / 2.58] | n/a | n/a | n/a |
| AD | 2.47 [2.27 / 2.66] | 2.47[a] [2.33 / 2.53] | 2.32 [2.10 / 2.49] | 2.49[a] [2.17 / 2.71] |
| ADp | 2.24[#] [2.04 / 2.31] | 2.40[, ab] [2.21 / 2.51] | 2.43 [2.34 / 2.51] | 2.14[#, ab] [1.96 / 2.35] |
| ADi | 2.17 [1.98 / 2.35] | 2.24[#, ab] [2.14 / 2.37] | 2.35[#] [2.21 / 2.52] | 2.38[#, a] [2.24 / 2.48] |
| CD | 2.27 [2.09 / 2.36] | 2.10[#, b] [1.87 / 2.33] | 2.14[#] [1.92 / 2.29] | 1.88[#, b] [1.75 / 2.09] |

Cholesterol concentrations are presented as medians and [25th / 75th] percentiles in mmol/L. [#■]Different symbols indicate significant differences within a row (time-point differences in a group). [ab]Lowercase letters indicate significant effects within a column (group differences at one timepoint). Significant differences are identified by P values ≤ 0.05 using repeated measures ANOVA with Tukey HSD. BL, baseline group (n = 8); AD, group fed atherogenic diet (n = 10); ADp, group fed atherogenic diet + pectin (n = 10); ADi, group fed atherogenic diet + inulin (n = 10); CD, group fed conventional diet (n = 10); n/a, not available.

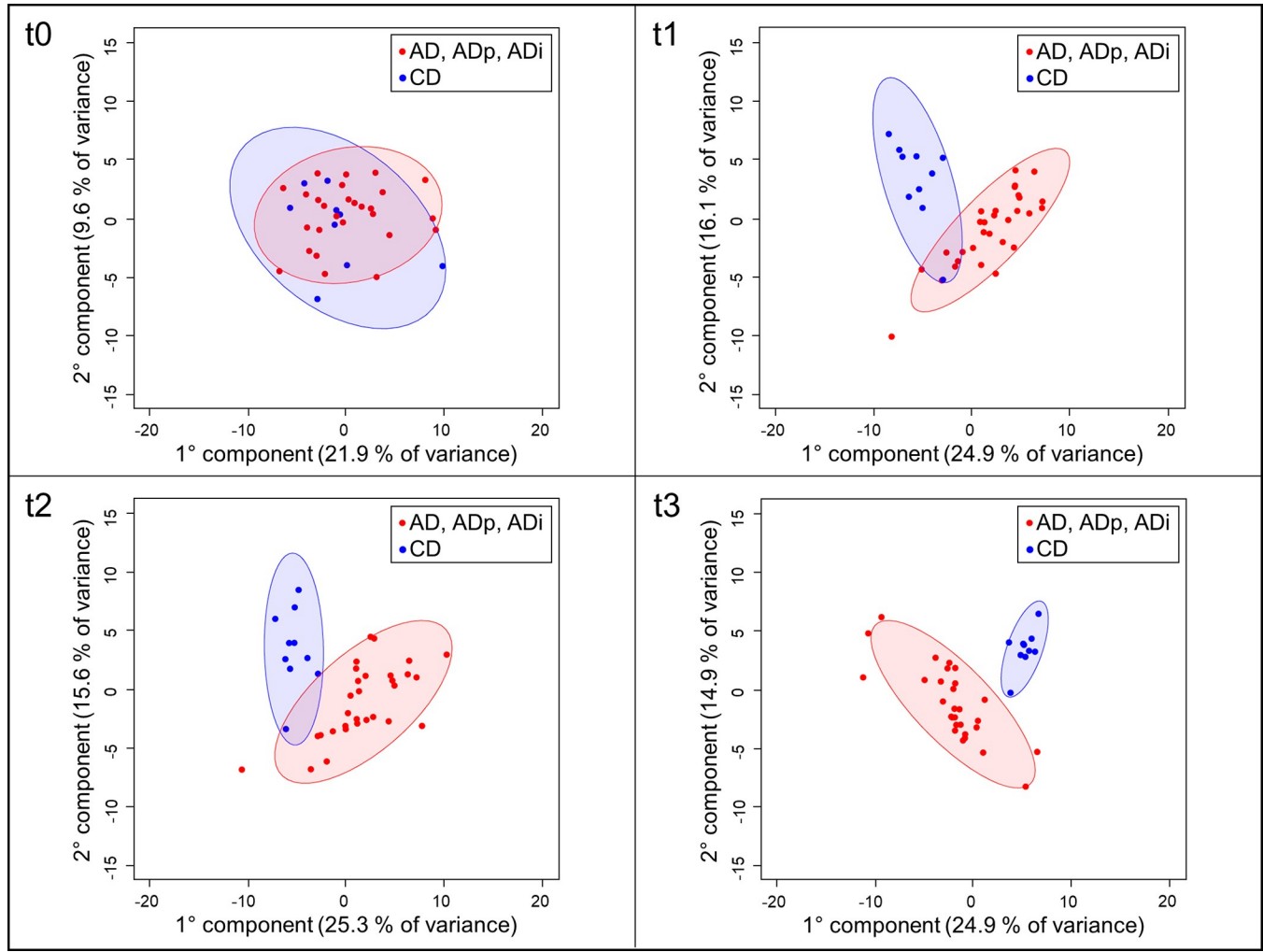

**Fig 2. PCA analysis of the ¹H NMR spectra of serum samples.** Data present the time point before the feed change (t0) and time points 1–3 months after the feed changes (t1–t3) for all feeding groups (AD, ADp, ADi, CD).

BCS and BFT did not significantly differ between the four feeding groups. One reason for this difference may be the incomplete digestion of fat, as fecal fat content increased after diet change (t1) in the groups fed the atherogenic diets.

Independent of supplementation with pectin and inulin, significant changes in blood parameters were observed after one month of feeding with the atherogenic diet. PCA showed a clear clustering into the groups fed the atherogenic diet (AD, ADp, Adi) and the conventionally fed group (CD) after the feed change (t1–t3). These effects were also observed in blood parameters such as TG, BA, LIPC and CHOL.

PLS-DA did not allow a clear metabolome-based distinction between the four feeding groups for t1 and t2 because of the small differences in circulating metabolites between the AD, ADp, and ADi groups. Additionally, at time t3, no separation was possible in the PLS-DA between the groups fed an atherogenic diet (AD, ADp, and ADi). These results were also supported by the blood levels of TG, BA, LIPC and CHOL at time t3, which did not show significant differences between these groups. Therefore, NMR spectroscopy for metabolome analyses has been demonstrated to be a powerful tool for expanding the density of data in veterinary

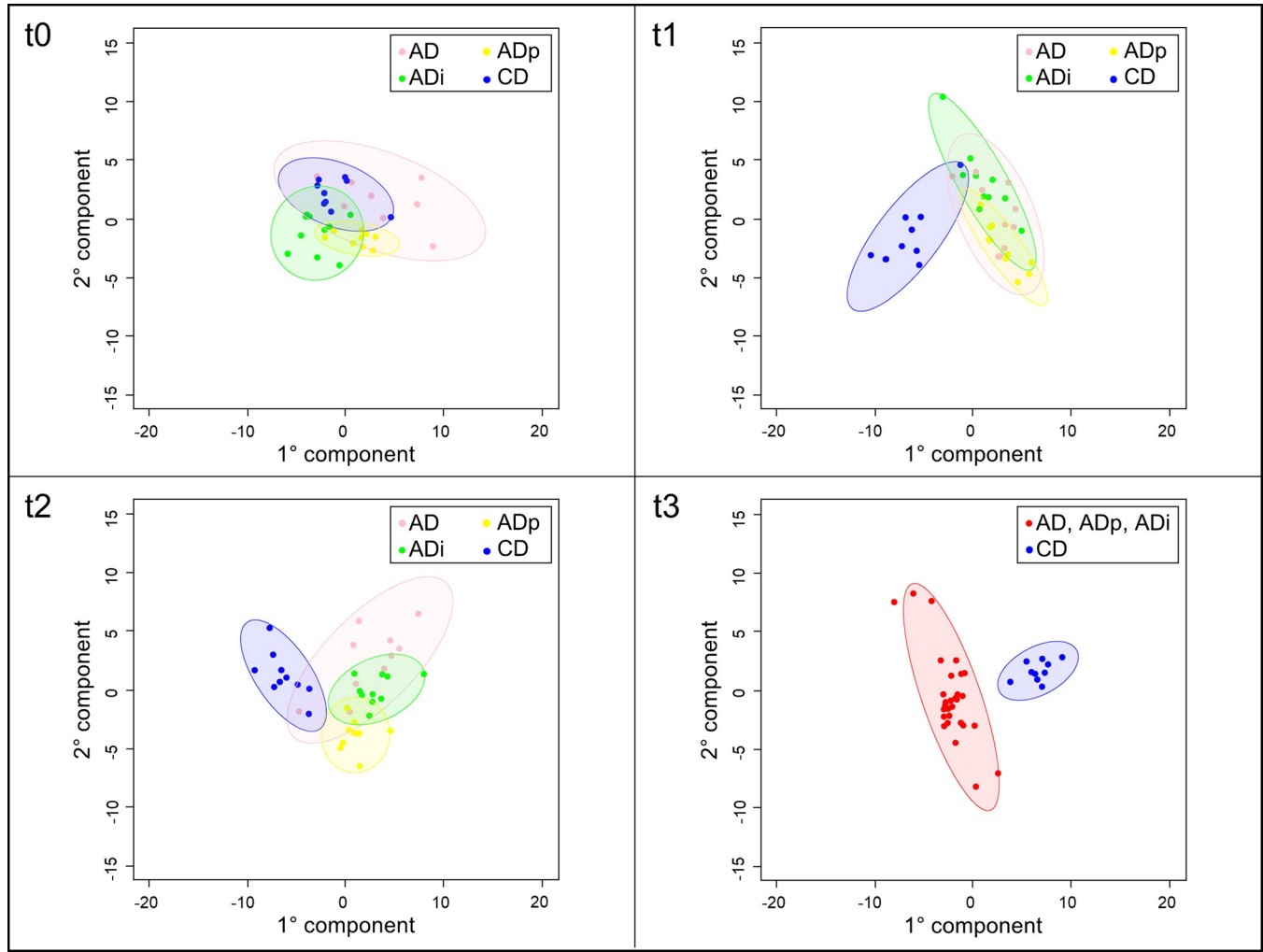

**Fig 3. PLS-DA analysis of ¹H NMR spectra of serum samples.** Data present the time point before the feed change (t0) and the time points 1–3 months after feed changes (t1–t3) for all feeding groups (AD, ADp, ADi, CD).

nutrition research and should be considered in future studies. However, our results might be somewhat confounded by the fact that the pigs were kept on wood shavings per ethical considerations, and the intake of wood shavings could not be excluded.

Interestingly, supplementation with pectin or inulin did not affect atherogenic diet-induced changes in blood parameters such as cholesterol. These findings are in contrast to those obtained by other authors, who reported beneficial effects of pectin on circulating cholesterol in pigs and humans [42,43,54,55]. It is tempting to speculate that 5% of pectin used in the current study was too low. Similar pectin levels have been used in other pig studies, showing cholesterol-lowering effects [43,54]. It is possible that the high-energy intake of pigs in the present study overlapped with the effects of pectin and inulin. Another factor could be the duration of the study, which lasted 15 weeks. Other studies have been conducted over a shorter feeding period (10–12 weeks), showing cholesterol-lowering effects for pectin [54] and lower BW gain for inulin [44] in pigs.

Interestingly, all groups, including the baseline group, developed moderate lesions in the coronary arteries. It can be speculated that the Saddleback pigs in our study may have a

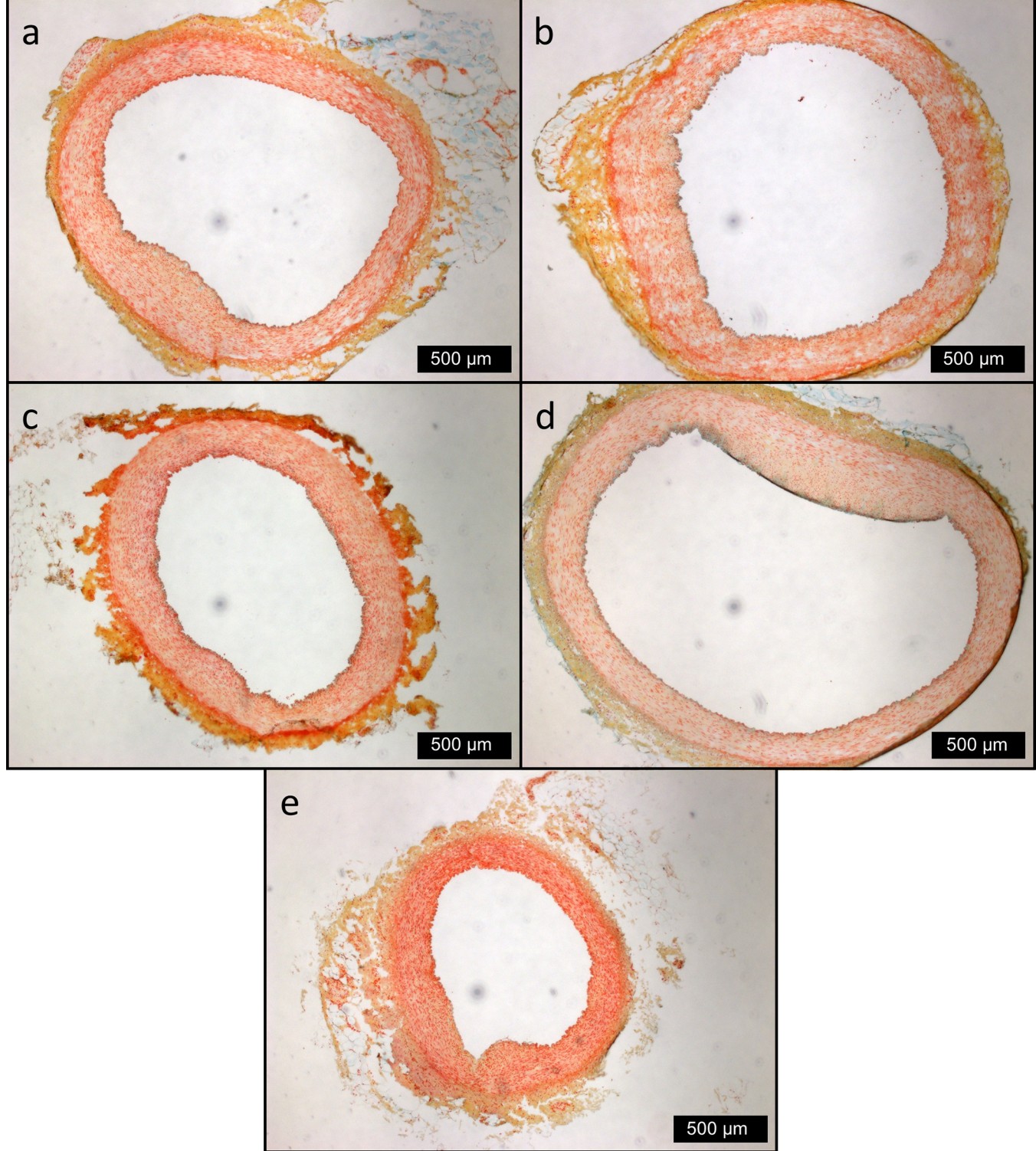

**Fig 4. Representative images of LAD cross-sections of all groups.** Movat's Pentachrome staining (frozen). Scale bar 500 μm. **a** BL. **b** AD. **c** ADp. **d** ADi. **e** CD.

**Table 6. Parameters of LAD plaque formation in all groups.**

| Group | Percentage of pigs within a group showing vascular lesions in LAD (%) | Plaque size in plaque-positive pigs ($\mu m^2$) |
|---|---|---|
| BL | 63[ab] | 35061 [22930 / 71273] |
| AD | 50[a] | 157062 [73901 / 198848] |
| ADp | 100[b] | 40207 [21287 / 181180] |
| ADi | 70[ab] | 128539 [34196 / 212237] |
| CD | 70[ab] | 46162 [14275 / 139486] |

Data show the plaque frequency per group (%) and plaque size ($\mu m^2$) of the plaque-positive pigs per group, expressed as medians and [25th / 75th] percentiles.

[ab]Lowercase letters indicate significant effects within the columns. Significant differences between the groups (BL, n = 8; AD, ADp, ADi, CD, n = 10) were identified using the chi-squared test (P ≤ 0.05). LAD, left anterior descending branch of the left coronary artery; BL, baseline group; AD, group fed atherogenic diet; ADp, group fed atherogenic diet + pectin; ADi, group fed atherogenic diet + inulin; CD, group fed conventional diet.

predisposition to plaque development. A familial disposition for plaque formation is also known for other pig breeds, such as Rapacz pigs [20,21]. However, the etiology in Rapacz pigs is hypercholesterolemia. In contrast, in this study, the cholesterol levels of all pigs remained within the normal range [56] despite significant increases in cholesterol after feeding the atherogenic diet. Beside these findings, it must be emphasized that the plaques represent very early lesion stages of atherosclerosis.

An unexpected finding was related to the higher lesion rate per group in the coronary arteries of the ADp group with pectin supplementation (10/10 animals affected) compared with the atherogenic diet without supplementation (5/10 animals affected). Interestingly, pectin supplementation induced smaller lesions in the coronary arteries than the atherogenic diet alone or the atherogenic diet supplemented with inulin. It remains unclear whether a higher lesion rate or smaller lesions are more beneficial. In contrast, the effect of inulin on plaque formation in coronary arteries was similar to that in pigs fed an atherogenic diet without fermentable carbohydrates. The plaques in the coronary arteries of the pectin-supplemented group were similar in size to those of the conventionally fed group. The baseline group, which was 15 weeks younger than the other groups, had the smallest plaques in their coronary arteries. Because the baseline group was slaughtered earlier, the younger age is probably the reason for the smallest plaques in this group. Therefore, age may be a critical factor in plaque formation in the coronary arteries. This indicates that Saddleback pigs are capable of developing vascular lesions within a comparatively young age. In the present study, the pigs were five months old at the beginning of the feeding study. However, it is possible that earlier or longer administration or higher doses of pectin and inulin would have exerted more beneficial effects on the vascular system and metabolism. In particular, to investigate the progression of lesion development, changes during a longer supplementation period should be the focus of future studies.

## Conclusion

Our findings indicate that Saddleback pig models are very useful for studying the effects of diets on atherosclerosis because these pigs develop spontaneous vascular lesions in their coronary arteries in early life. In contrast to our hypothesis, pectin and inulin supplementation of

an atherogenic diet did not show any major beneficial effects on atherosclerosis and cardiome-tabolic risk factors under the applied conditions. In human medicine, in addition to supplementation with pectin, inulin, or other fermentable carbohydrates, energy restriction is indispensable.

## Supporting information

**S1 File. ARRIVE guideline.**
(DOCX)

**S1 Table. Original data of BW, BCS and BFT.**
(XLSX)

**S2 Table. Serum parameters of liver and lipid metabolism.**
(DOCX)

**S3 Table. Original data of serum analysis.**
(XLSX)

**S4 Table. Original data of plaque formation in the LAD.**
(XLSX)

## Acknowledgments

The authors are grateful to Jana Tietke and Beatrice Ladanyi for providing technical support and to Sabine Kleemann and Michael Wacker for taking care of the pigs. The authors also thank Claudia Wiacek, Susann Kauschke, Elena Theiner, Clara Müller, Louisa Esmeralda Maria Sroka, Janine Starzonek, and Niklas Fuchs for supporting sample collection and slaughtering and agro Food Solution GmbH and Beneo GmbH for providing pectin and inulin.

## Author Contributions

**Conceptualization:** Lisa Wahl, Gabriele I. Stangl, Ingrid Vervuert.

**Data curation:** Lisa Wahl, Melina Raschke.

**Formal analysis:** Lisa Wahl, Melina Raschke, Johannes Wittmann.

**Funding acquisition:** Gabriele I. Stangl, Ingrid Vervuert.

**Investigation:** Lisa Wahl, Melina Raschke, Armin Regler, Corinna Brandsch, Ingrid Vervuert.

**Methodology:** Lisa Wahl, Melina Raschke, Johannes Wittmann, Armin Regler, Steffen Heelemann, Corinna Brandsch, Gabriele I. Stangl, Ingrid Vervuert.

**Project administration:** Ingrid Vervuert.

**Resources:** Steffen Heelemann, Gabriele I. Stangl, Ingrid Vervuert.

**Supervision:** Ingrid Vervuert.

**Visualization:** Lisa Wahl.

**Writing – original draft:** Lisa Wahl.

**Writing – review & editing:** Lisa Wahl, Johannes Wittmann, Armin Regler, Steffen Heelemann, Corinna Brandsch, Gabriele I. Stangl, Ingrid Vervuert.

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
