## [Decision Letter · Decision Letter 0]

30 May 2022

PONE-D-22-07717Effects of atherogenic diet supplemented with fermentable carbohydrates on metabolic responses and plaque formation in coronary arteries using a Saddleback pig modelPLOS ONE

Dear Dr. Vervuert,

Thank you for submitting your manuscript to PLOS ONE. After careful consideration, we feel that it has merit but does not fully meet PLOS ONE’s publication criteria as it currently stands. Therefore, we invite you to submit a revised version of the manuscript that addresses the points raised during the review process.

We look forward to receiving your revised manuscript.

Kind regards,

Karin Jandeleit-Dahm

Academic Editor

PLOS ONE

Journal Requirements:

Additional Editor Comments (if provided):

There are concerns about the model, statistical analysis, controls and the assessment of plaque morphology and composition.

Please assess the reviewer comments and provide a strong rationale for the model, present data in form of additional figures or tables and perform additional assessment of plaque morphology and composition.

Reviewers' comments:

Reviewer's Responses to Questions

**Comments to the Author**

1. Is the manuscript technically sound, and do the data support the conclusions?

Reviewer #1: Partly

Reviewer #2: Partly

2. Has the statistical analysis been performed appropriately and rigorously? 

Reviewer #1: Yes

Reviewer #2: Yes

3. Have the authors made all data underlying the findings in their manuscript fully available?

Reviewer #1: Yes

Reviewer #2: Yes

4. Is the manuscript presented in an intelligible fashion and written in standard English?

Reviewer #1: Yes

Reviewer #2: Yes

5. Review Comments to the Author

Reviewer #1: The current manuscript describes a porcine model of coronary artery atherosclerosis and the effects of the fermentable carbohydrates, pectin or inulin on disease after 15 weeks feeding with a Western-style diet. While the Western-style diet increased serum lipid levels, surprisingly no apparent change in the incidence of atherosclerotic lesions in the LAD was noted compared to the control diet group, although lesion size appeared to increase in the atherogenic diet group. This raises questions on the viability of this experimental model for the testing potential anti-atherogenic interventions. Also, key information is missing regarding the measurement of atherosclerosis in the LAD artery including at what anatomical site/region of the LAD artery was atherosclerosis incidence and lesion size measured and was this site/region uniform for all measurements across the different animals. Also, it would be helpful for lesion size to be expressed in the more conventional mean +/- SD format and shown as a scatter plot with the individual values of lesion size for each animal provided, as well as an accompanying stats analysis. It is also important for further analysis to be performed regarding the stage and phenotype of the lesions formed. For example, are the lesion early fatty streaks or have they progressed to a more advanced state. Therefore, histological assessment of macrophage, collagen, SMC and other indices of lesion status should be performed. While lesion size is one important measurement, lesion inflammatory status and stability are also critical readouts when assessing the utility of this porcine model of atherosclerosis. If as suspected, the lesions are early in nature, then perhaps 15 weeks atherogenic diet is insufficient time to drive the development of advanced atherosclerotic lesions in a larger animal such as the Saddleback pig and hence it is difficult to ascertain the impact of potential anti-atherogenic interventions such as pectin and inulin. Have the authors performed prior studies examining the impact of the time of feeding of an atherogenic diet on atherosclerosis development in this model? If not this appears a sensible first step in model development and validation.

Several other aspects warrant attention:

1. What is the rationale for the choice of 5% pectin and inulin?

2. What was the rationale for 15 weeks of atherogenic diet? This appears short for a larger animal such as a pig.

3. The large tables of data are difficult to analyze and digest the data. The additional use of figures to show key data sets would be beneficial.

4. A rationale for the choice of Saddleback pigs would be beneficial. How does this model compare to other porcine models of atherosclerosis?

Reviewer #2: Major Comments:

1) The Introduction should highlight clinical trials in humans that have investigated fermentable carbohydrates and cardiovascular risk (eg Bocheng Xu et al, 2022).

2) A further detailed investigation into the atherogenic plaques need to be performed. Are the plaques stable/unstable? This can be easily done by immunohistochemistry investigating smooth muscle cell content, macrophage/immune cells, and fibrosis?

3) A more detailed analysis on why pectin was more effective in reducing plaque than inulin? The hypothesis is not supported in the study and needs further clarification.

Minor Comments:

1) In the introduction line 57, "provoked" plaques is not the right term to use, please ammend.

2) Statistical significances in the study in particular Table 5 is relatively confusing as to which group or timepoint it refers to. Please explicitly state significance values and group compared to.

6. PLOS authors have the option to publish the peer review history of their article (what does this mean?). If published, this will include your full peer review and any attached files.

Reviewer #1: No

Reviewer #2: No

---

## [Author Response · Author response to Decision Letter 0]

18 Jul 2022

Response letter

“Effects of atherogenic diet supplemented with fermentable carbohydrates on metabolic responses and plaque formation in coronary arteries using a Saddleback pig model”

Manuscript Submission No.: PONE-D-22-07717

Summary

We would like to thank the Associate Editor and the Reviewers for a thorough analysis of the submitted work as well as precise and constructive comments which helped to better organize and polish the manuscript.

These contributions were expressed more clearly throughout the manuscript via:

• additional figures to illustrate the results of serum parameter analyses

• a more detailed description of the coronary artery sampling procedure

• additional information and further investigations on the phenotype of vascular lesions

• information about the statistical evaluation of plaque size

1) Response to the Editor`s Report

Comment 1: Please ensure that your manuscript meets PLOS ONE's style requirements, including those for file naming. The PLOS ONE style templates can be found at https://journals.plos.org/plosone/s/file?id=wjVg/PLOSOne_formatting_ sample_main_body.pdf and https://journals.plos.org/plosone/s/file?id=ba62/ PLOSOne_formatting_sample_title_authors_affiliations.pdf

Response: Thank you for this comment. We checked that our manuscript meets PLOS ONE's style requirements by using your English Editing Service to improve the manuscript.

Comment 2: To comply with PLOS ONE submissions requirements, in your Methods section, please provide additional information regarding the experiments involving animals and ensure you have included details on (1) methods of sacrifice, (2) methods of anesthesia and/or analgesia, and (3) efforts to alleviate suffering.

Response: Thank you very much for this response. The experiment was approved by the Ethics Committee for Animal Rights Protection of the Leipzig District Government (no. TVV 04/20) in accordance with German legislation for animal rights and welfare (line 114−117). The licensing authority also considered the burden on the animals to be low due to the blood and fecal samples taken once a month during the experiment. All methods such as blood sampling, fecal sampling and slaughtering has been described in the chapter “materials and methods” (line 156−188). During the experiment, great importance was attached to animal welfare. In addition to occupation with the bedding material (wood shavings), the pigs were provided with new organic manipulable material twice a week like chewing ropes, biting woods and a stimulus rod.

In addition, strict inclusion and exclusion criteria were established prior to the study, which are described as follows in supplementary file 1 (S1 File. ARRIVE guideline):

Including criteria:

- Breed: Saddle back pigs (5 litters, same sire)

- Age: 5 Month

- Health status: Healthy (examined by clinical examination and blood check)

Excluding criteria:

- A score sheet was developed to establish the criteria for excluding animals from the study. The score was developed specifically for pigs and modified based on the results for pain assessment in pigs by Ison et al. (2016).

- Scored parameter: general behaviour, condition score, feed and water intake, quality of faeces, breathing rate, body temperature, limbs, injuries (especially tail, ears and flanks), umbilical hernia, symptoms of gastric ulcera

- Score > 0 to 1 is reached for one parameter � animals were intensively observed (clinical examination three times a day)

- Score > 1 is reached for more than one parameter or a score of 3 is reached for one parameter � the animals are excluded of the study and undergo further diagnostic examinations (e.g. further lameness diagnostics) and, if necessary, lege artis medical treatment (e.g. pain therapy or antibiosis).

- If a complete recovery of the animals is possible, they are included in the trial again. 

- If individual animals cannot be recovered and included in the trial, these animals are euthanised to avoid further pain and suffering.

As we described in the manuscript, the pigs were slaughtered by a qualified butcher in accordance with European and German law [Council Regulation (EC) No 1099/2009 of 24 September 2009, Tierschutz-Schlachtverordnung, § 4 Tierschutzgesetz] (line 173−177). We added a detailed description of the slaughtering process as follows in supplementary file 1 (S1 File. ARRIVE guideline):

Slaughtering process

Pigs were adapted and familiar with manual handling. All animals underwent a veterinary examination immediately before slaughtering and all pigs were diagnosed to be clinically healthy. Slaughter of the pigs took place on the same farm where the pigs were kept during the experiment (distance from the stable to the slaughterhouse: 200 m), which excluded long transportation stress. The pigs were individually stunned with an electric stunning system according to European and German law [Council Regulation (EC) No 1099/2009 of 24 September 2009, Tierschutz-Schlachtverordnung, § 4 Tierschutzgesetz] (TGB 200; Hubert Haas, Neuler, Germany, brain and brain-heart perfusion, minimum current of 1.3 A within the 1st second, 250 V, alternating current with 50−100 Hz, perfusion duration with 1.3 A for at least 4 seconds, data were recorded by the stunning system) by the qualified butcher and then sacrificed by blood withdrawal within 10 seconds (knife with a blade length of at least 12 cm, stabbing direction in the jugular fossa 2−3 cm in front of the sternal apex in the direction of the opposite scapula, cut length of 2−3 cm, opening of the brachiocephalic trunk and jugular vein, blood loss of at least 3−4 L in 30 seconds). Thereafter, the carcasses were examined by an official veterinarian and declared edible. The carcasses were processed by regional butchers.

We hope that we have considered your comments in an adequate way.

Comment 3: There are concerns about the model, statistical analysis, controls and the assessment of plaque morphology and composition. Please assess the reviewer comments and provide a strong rationale for the model, present data in form of additional figures or tables and perform additional assessment of plaque morphology and composition.

Response: We appreciate the effort of the associate editor as well as of the reviewers for highlighting flaws in our submission. The comments and concerns associated to the review have been carefully inspected by us. Based on the review, the manuscript has been refined and adapted. We added a figure for each of the serum parameters (Triglycerides [TG], hepatic triglyceride lipase [LIPC], bile acids [BA], and cholesterol [CHOL] line 336−343). As the visualization of the statistic is complex, we divided Table 5 into sub tables (5a-5d, line 345−387) to provide information about significances. 

2) Response to Reviewer 1

Comment 1: The current manuscript describes a porcine model of coronary artery atherosclerosis and the effects of the fermentable carbohydrates, pectin or inulin on disease after 15 weeks feeding with a Western-style diet. While the Western-style diet increased serum lipid levels, surprisingly no apparent change in the incidence of atherosclerotic lesions in the LAD was noted compared to the control diet group, although lesion size appeared to increase in the atherogenic diet group. This raises questions on the viability of this experimental model for the testing potential anti-atherogenic interventions.

Response: The authors confirm that the lesions were only very moderate. At this lesion stage, the plaques were of minor significance in terms of health consequences for the pigs. Nevertheless, the authors consider the formation of spontaneous plaques without the induction by feeding cholesterol and cholic acids relevant to mention. Although of minor significance, group-related differences were detected in plaque formation.

The influencing aspects such as duration of feeding period, levels of pectin and inulin in the diet as well as age of the pigs were addressed in the discussion as follows: “Because the baseline group was slaughtered earlier, the younger age is probably the reason for the smallest plaques in this group. Therefore, age may be a critical factor in plaque formation in the coronary arteries. This indicates that Saddleback pigs are capable of developing vascular lesions within a comparatively young age. In the present study, the pigs were five months old at the beginning of the feeding study. However, it is possible that earlier or longer administration or higher doses of pectin and inulin would have exerted more beneficial effects on the vascular system and metabolism. In particular, to investigate the progression of lesion development, changes during a longer supplementation period should be the focus of future studies.” (line 517−526)

Comment 2: Also, key information is missing regarding the measurement of atherosclerosis in the LAD artery including at what anatomical site/region of the LAD artery was atherosclerosis incidence and lesion size measured and was this site/region uniform for all measurements across the different animals.

Response: The arterial section for the histological analyses was obtained from the same site of each animal. The section was excised 3 mm after the bifurcation of the left main stem of the coronary artery, where the artery gets divided into the left circumflex (LCX) and the left anterior descending (LAD). From each animal four 2 mm segments of the LAD were gained. The 1st and 3rd segments were fixed to prepare the PEG sections; the 2nd and 4th segments were snap frozen for the preparation of the frozen sections. We included a more detailed description in the manuscript (“materials and methods”, “slaughtering and sampling of coronary arteries”, line 180−188).

Comment 3: Also, it would be helpful for lesion size to be expressed in the more conventional mean +/- SD format and shown as a scatter plot with the individual values of lesion size for each animal provided, as well as an accompanying stats analysis.

Response: As data of lesion size were not normally distributed, the authors decided to use median and [25th / 75th] percentile to present the data. However, the expression in mean ± SD does not change the outcome. For statistical evaluation of plaque size Kruskal-Wallis test with Bonferroni correction was used (P = 0.33). We included this information in the manuscript (“results”, “plaque formation”, line 433−434). As there were no significant group differences in plaque size, the authors decided to present only the plaque sizes of the plaque-positive pigs. We provided the single values in a supplementary file (S5 Table. Original data of plaque formation in the LAD).

Comment 4: It is also important for further analysis to be performed regarding the stage and phenotype of the lesions formed. For example, are the lesion early fatty streaks or have they progressed to a more advanced state. Therefore, histological assessment of macrophage, collagen, SMC and other indices of lesion status should be performed. While lesion size is one important measurement, lesion inflammatory status and stability are also critical readouts when assessing the utility of this porcine model of atherosclerosis.

Response: We agree that lesion inflammatory status and plaque stability are critical readouts to evaluate the progress of atherogenesis. However, all lesions detected in the LAD sections represented very early lesion stages of atherosclerosis. None of the pigs had advanced lesions with fibrous caps or lipid cores. Additionally, the area of atherosclerotic lesions in relation to the area of arterial tissue was on average below 6% in all groups of pigs, which confirm that the atherosclerotic changes in the blood vessel were moderate intima thickening. Moreover, staining of the LAD sections with Oil red illustrates that only one pig had detectable amounts of lipids in the lesions. Analysis of Von Kossa staining which usually visualize vascular calcification, revealed no calcification spots in the LAD of pigs. As suggested by the reviewers, we added additional immunohistochemical analyses. First, we tried to record macrophages by using a specific mouse anti pig macrophages antibody (MCA2317GA, BIO-RAD), but could not detect any macrophages in the LAD lesions. Staining of smooth muscle cells performed with a mouse anti human actin alpha (smooth muscle) antibody (MCA5781GA, BIO-RAD), which was used for porcine tissue before, showed stainable smooth muscle cells but no differences between the feeding groups. Based on the current data, we suppose that feeding pigs for 15 weeks with an atherogenic diet can only induce very early forms of atherosclerotic lesions. We therefore conclude that a longer feeding period is required for the development of more pronounced vascular lesions with a detectable inflammatory status. Authors added the respective information in the manuscript (“material and methods” line 266−272, “results” line 429−431 and “discussion” line 504−506).

Comment 5: If as suspected, the lesions are early in nature, then perhaps 15 weeks atherogenic diet is insufficient time to drive the development of advanced atherosclerotic lesions in a larger animal such as the Saddleback pig and hence it is difficult to ascertain the impact of potential anti-atherogenic interventions such as pectin and inulin.

Response: The authors agree with this comment. We discussed this point, especially the duration of the experiment, in the discussion line 517−526 (see comment 1 above). 

Comment 6: Have the authors performed prior studies examining the impact of the time of feeding of an atherogenic diet on atherosclerosis development in this model? If not this appears a sensible first step in model development and validation.

Response: Thank you for this comment. This was one idea of the experiment. We want to use this study to be the basis of further research.

Comment 7: What is the rationale for the choice of 5% pectin and inulin?

Response: The used dosages or lower dosages for pectin and inulin were already fed in other studies in pigs and the chosen levels have been already linked to positive effects [36,43,44,54]. We added this information in the manuscript (“materials and methods”, “feed management”, line 132−133)

Comment 8: What was the rationale for 15 weeks of atherogenic diet? This appears short for a larger animal such as a pig.

Response: The authors acknowledged this comment. The duration is an issue of concern. The experiment was planned for a longer time period, starting the experiment with 5-month-old pigs and mean body weight of 97.5 kg. Because of the high body weight increase during the experiment, pigs started to change their behavior in the 14th week of the experiment by decreasing their activity level. 

Therefore, it was not possible to use these pigs for a longer time period due to reasons of animal welfare. Furthermore, it could not be excluded that these changes in activity level may have an impact on the outcome. Consequently, we decided to stop the experiment after 15 weeks. Authors would like to emphasize that all pigs already reached a BCS between 4 and 5 on a scale 0 to 5 at this timepoint.

Comment 9: The large tables of data are difficult to analyze and digest the data. The additional use of figures to show key data sets would be beneficial.

Response: We added a figure for each serum parameter (Triglycerides [TG], hepatic triglyceride lipase [LIPC], bile acids [BA], and cholesterol [CHOL]). As you also mentioned, the presentation of the statistic is complex. Therefore, we provide tables to give information about significances. However, it is very difficult to show all P values separately. We hope that by providing data in sub tables (line 345−387) and figures (Fig 1: Triglycerides, bile acids, hepatic triglyceride lipase, and cholesterol concentrations in serum, line 336−343) data presentation becomes clearer.

Comment 10: A rationale for the choice of Saddleback pigs would be beneficial. How does this model compare to other porcine models of atherosclerosis?

Response: We compared the commonly used porcine models of atherosclerosis in the introduction (line 65−72). Previous studies used cholesterol and cholic acids to induce the formation of atherosclerotic plaques, familiar predisposed pigs or genetically modified animals [17−22]. However, it is also known that pigs can develop plaques spontaneously [23−24]. Authors would like to emphasize that we want to use pigs which can develop plaques spontaneously to get deeper insight about the consequences of consuming typical western style diets. 

3) Response to Reviewer 2

Comment 1: The Introduction should highlight clinical trials in humans that have investigated fermentable carbohydrates and cardiovascular risk (eg Bocheng Xu et al, 2022).

Response: Thank you for this information. Indeed, we did not acknowledge this paper. However, we have included the information about the interesting meta-analysis of Xu et al. [30] in the introduction (line 77−78). 

Comment 2: A further detailed investigation into the atherogenic plaques need to be performed. Are the plaques stable/unstable? This can be easily done by immunohistochemistry investigating smooth muscle cell content, macrophage/immune cells, and fibrosis?

Response: The reviewers are completely right, that lesion inflammatory status and plaque stability are critical readouts to evaluate the progress of atherogenesis. However, all lesions detected in the LAD sections represented very early lesion stages of atherosclerosis. None of the pigs had advanced lesions with fibrous caps or lipid cores. Additionally, the area of atherosclerotic lesions in relation to the area of arterial tissue was on average below 6% in all groups of pigs, which confirm that the atherosclerotic changes in the blood vessel were moderate intima thickening. Moreover, staining of the LAD sections with Oil red illustrates that only one pig had detectable amounts of lipids in the lesions. Analysis of Von Kossa staining which usually visualize vascular calcification, revealed no calcification spots in the LAD of pigs. As suggested by the reviewers, we added additional immunohistochemical analyses. First, we tried to record macrophages by using a specific mouse anti pig macrophages antibody (MCA2317GA, BIO-RAD), but could not detect any macrophages in the LAD lesions. Staining of smooth muscle cells performed with a mouse anti human actin alpha (smooth muscle) antibody (MCA5781GA, BIO-RAD), which was used for porcine tissue before, showed stainable smooth muscle cells but no differences between the feeding groups. Based on the current data, we suppose that feeding pigs for 15 weeks with an atherogenic diet can only induce very early forms of atherosclerotic lesions. We therefore conclude that a longer feeding period is required for the development of more pronounced vascular lesions with a detectable inflammatory status. Authors added the respective information in the manuscript (“material and methods” line 266−272, “results” line 429−431 and “discussion” line 504−506).

Comment 3: A more detailed analysis on why pectin was more effective in reducing plaque than inulin? The hypothesis is not supported in the study and needs further clarification.

Response: The authors speculated that the production of short chain fatty acids (SCFA) in the large intestine by adding pectin or inulin to the diet may play a significant role. The analyses of SCFA are currently under investigation.

Comment 4: In the introduction line 57, "provoked" plaques is not the right term to use, please ammend.

Response: The authors agree to your point. We have changed the word "provoked" into "induced" in the introduction (line 57).

Comment 5: Statistical significances in the study in particular Table 5 is relatively confusing as to which group or timepoint it refers to. Please explicitly state significance values and group compared to.

Response: 

We added a figure for each serum parameter (Triglycerides [TG], hepatic triglyceride lipase [LIPC], bile acids [BA], and cholesterol [CHOL]). As you also mentioned, the presentation of the statistic is complex. Therefore, we provide tables to give information about significances. However, it is very difficult to show all P values separately. We hope that by providing data in sub tables (line 345−387) and figures (Fig 1: Triglycerides, bile acids, hepatic triglyceride lipase, and cholesterol concentrations in serum, line 336−343) data presentation becomes clearer.

---

## [Decision Letter · Decision Letter 1]

12 Sep 2022

Effects of atherogenic diet supplemented with fermentable carbohydrates on metabolic responses and plaque formation in coronary arteries using a Saddleback pig model

PONE-D-22-07717R1

Dear Dr. Vervuert,

We’re pleased to inform you that your manuscript has been judged scientifically suitable for publication and will be formally accepted for publication once it meets all outstanding technical requirements.

Kind regards,

Karin Jandeleit-Dahm

Academic Editor

PLOS ONE

Additional Editor Comments (optional):

Concerns have been addressed.

Reviewers' comments:

Reviewer's Responses to Questions

**Comments to the Author**

1. If the authors have adequately addressed your comments raised in a previous round of review and you feel that this manuscript is now acceptable for publication, you may indicate that here to bypass the “Comments to the Author” section, enter your conflict of interest statement in the “Confidential to Editor” section, and submit your "Accept" recommendation.

Reviewer #1: All comments have been addressed

2. Is the manuscript technically sound, and do the data support the conclusions?

Reviewer #1: Partly

3. Has the statistical analysis been performed appropriately and rigorously? 

Reviewer #1: Yes

4. Have the authors made all data underlying the findings in their manuscript fully available?

Reviewer #1: Yes

5. Is the manuscript presented in an intelligible fashion and written in standard English?

Reviewer #1: Yes

6. Review Comments to the Author

Reviewer #1: (No Response)

7. PLOS authors have the option to publish the peer review history of their article (what does this mean?). If published, this will include your full peer review and any attached files.

Reviewer #1: No

---

## [Editor Report · Acceptance letter]

27 Sep 2022

PONE-D-22-07717R1 

Effects of atherogenic diet supplemented with fermentable carbohydrates on metabolic responses and plaque formation in coronary arteries using a Saddleback pig model 

Dear Dr. Vervuert:

I'm pleased to inform you that your manuscript has been deemed suitable for publication in PLOS ONE. Congratulations! Your manuscript is now with our production department. 

Kind regards, 

on behalf of

Professor Karin Jandeleit-Dahm 

Academic Editor

PLOS ONE